# Ventilator-Associated Pneumonia in COVID-19 Patients Admitted in Intensive Care Units: Relapse, Therapeutic Failure and Attributable Mortality—A Multicentric Observational Study from the OutcomeRea Network

**DOI:** 10.3390/jcm12041298

**Published:** 2023-02-06

**Authors:** Paul-Henri Wicky, Claire Dupuis, Charles Cerf, Shidasp Siami, Yves Cohen, Virginie Laurent, Bruno Mourvillier, Jean Reignier, Dany Goldgran-Toledano, Carole Schwebel, Stéphane Ruckly, Etienne de Montmollin, Niccolò Buetti, Jean-François Timsit

**Affiliations:** 1Medical and Infectious Diseases Intensive Care Unit, Bichat Hospital, AP-HP, Paris Cité University, 46 rue Henri Huchard, 75018 Paris, France; 2UMR 1137, IAME, Université Paris Cité, 75018 Paris, France; 3Medical Intensive Care Unit, University Hospital Gabriel Montpied, 63000 Clermont-Ferrand, France; 4Polyvalent Intensive Care Unit, Hôpital Foch, 92150 Suresnes, France; 5General Intensive Care Unit, Sud Essonne Hospital, 91150 Etampes, France; 6Intensive Care Unit, University Hospital Avicenne, AP-HP, 93000 Bobigny, France; 7Polyvalent Intensive Care Unit, André Mignot Hospital, 78150 Le Chesnay, France; 8Medical Intensive Care Unit, University Hospital of Reims, 51100 Reims, France; 9Medical Intensive Care Unit, University Hospital of Nantes, 44000 Nantes, France; 10Medical and Surgical Intensive Care, Montfermeil Hospital, 93370 Montfermeil, France; 11Medical Intensive Care Unit, University Hospital Grenoble-Alpes, 38000 Grenoble, France; 12Infection Control Program and WHO Collaborating Centre on Patient Safety, Faculty of Medicine, University of Geneva Hospitals, 1205 Geneva, Switzerland

**Keywords:** COVID-19, ventilator-associated pneumonia, relapse, treatment failure, mortality

## Abstract

**Introduction:** Ventilator-associated pneumonia (VAP) incidence is high among critically ill COVID-19 patients. Its attributable mortality remains underestimated, especially for unresolved episodes. Indeed, the impact of therapeutic failures and the determinants that potentially affect mortality are poorly evaluated. We assessed the prognosis of VAP in severe COVID-19 cases and the impact of relapse, superinfection, and treatment failure on 60-day mortality. **Methods:** We evaluated the incidence of VAP in a multicenter prospective cohort that included adult patients with severe COVID-19, who required mechanical ventilation for ≥48 h between March 2020 and June 2021. We investigated the risk factors for 30-day and 60-day mortality, and the factors associated with relapse, superinfection, and treatment failure. **Results:** Among 1424 patients admitted to eleven centers, 540 were invasively ventilated for 48 h or more, and 231 had VAP episodes, which were caused by *Enterobacterales* (49.8%), *P. aeruginosa* (24.8%), and *S. aureus* (22%). The VAP incidence rate was 45.6/1000 ventilator days, and the cumulative incidence at Day 30 was 60%. VAP increased the duration of mechanical ventilation without modifying the crude 60-day death rate (47.6% vs. 44.7% without VAP) and resulted in a 36% increase in death hazard. Late-onset pneumonia represented 179 episodes (78.2%) and was responsible for a 56% increase in death hazard. The cumulative incidence rates of relapse and superinfection were 45% and 39.5%, respectively, but did not impact death hazard. Superinfection was more frequently related to ECMO and first episode of VAP caused by non-fermenting bacteria. The risk factors for treatment failure were an absence of highly susceptible microorganisms and vasopressor need at VAP onset. **Conclusions:** The incidence of VAP, mainly late-onset episodes, is high in COVID-19 patients and associated with an increased risk of death, similar to that observed in other mechanically ventilated patients. The high rate of VAP due to difficult-to-treat microorganisms, pharmacokinetic alterations induced by renal replacement therapy, shock, and ECMO likely explains the high cumulative risk of relapse, superinfection, and treatment failure.

## 1. Introduction

In patients admitted in an intensive care unit (ICU) and ventilated for Acute Respiratory Distress Syndrome (ARDS) due to SARS-CoV-2, ventilator-associated pneumonia (VAP) is a common complication, ranging from 30% to 86% of patients [1,2]. The cumulative incidence among COVID-19 patients can reach approximately 35/1000 ventilator days in European cohorts, which is 50% to 80% higher than in ARDS from other etiologies [3,4]. Several mechanisms explain the higher incidence of these superinfections [5,6]. They involve mucus plugs favoring atelectasis, diffuse alveolar damages, viral-induced immunoparalysis, impaired lung perfusion due to endothelial dysfunction, altered coagulation, and immunothrombosis [7,8]. Of note, the risk truly induced by corticosteroids or other immunomodulator therapies remains debated [9].

During the first pandemic wave, the fatality rate among COVID-19 patients reached 28.3% in ICU patients and 43% in ventilated patients [10]. The REA-REZO network suggested that the fraction of mortality due to VAP is higher in patients mechanically ventilated for severe COVID-19 compared to the general ICU population [2]. In addition, Giacobbe et al. estimated the attributable mortality of VAP was about 50%, and patients showed a doubling in length of stay [11]. The rate of VAP recurrences varied from 23% to 37% according to studies comparing viral and non-viral ARDS [12,13,14]. Moreover, treatment failures occurred in up to 66% of COVID-19 patients under extracorporeal membrane oxygenation (ECMO) [15], and severe complications, such as lung abscesses, were frequently diagnosed [16,17]. Still, those studies did not explore how far VAP treatment failures might have affected mortality.

The fraction of mortality explained by the severity of respiratory status, the increase in VAP incidence, and the consequences of difficult-to-treat superinfections remain to be evaluated. The available data could neither evaluate the risk factors for relapse, superinfection, and therapeutic failure, nor establish any relationship between therapeutic failure rates and mortality related to VAP.

The purpose of our study, which was conducted in a multicenter cohort that prospectively collected data, was to evaluate the attributable 60-day mortality of VAP. Moreover, we aimed to assess the incidence of and risk factors for treatment failure, relapse, and superinfection.

## 2. Methods

### 2.1. Study Design

We conducted a retrospective analysis using the French prospective multicenter OutcomeRea^TM^ database, where we selected patients from eleven French ICUs belonging to the OutcomeRea^TM^ network, who were admitted for severe COVID-19 and at risk of VAP.

### 2.2. Study Population

Every adult who was admitted in one of the defined ICUs between March 2020 and June 2021 with ARDS due to severe COVID-19 pneumonia was potentially eligible. Patients intubated for at least 48 h were considered at risk for VAP and, thus, included in our study. Patients were excluded if they were referred from another center or if a life-sustaining therapy withdrawal was expected within 48 h.

### 2.3. Definitions and Study Procedures

#### 2.3.1. Ventilator-Associated Pneumonia

The presence or absence of VAP was confirmed according to current guidelines; it required a positive quantitative culture of lower respiratory tract samples collected as recommended (bronchoalveolar lavage ≥10^4^ CFU/mL, plugged telescoping catheter ≥10^3^ CFU/mL, and endotracheal aspirate, ≥10^6^ CFU/mL) [18]. In patients with diffuse bilateral abnormalities on chest X-rays, new and progressive infiltrates were difficult to ascertain. An alteration in oxygenation and an increased need of vasopressors were considered to be indicating suspected VAP. An antibiotic treatment was considered adequate if the use of one or more antibiotics empirically initiated for VAP was active against the causative microorganism, and if the treatment was administered within the first 24 h after the VAP occurrence [19]. Early-onset VAP was defined as VAP occurring within the first 7 days after the initiation of mechanical ventilation, whereas late-onset was defined as VAP occurring after this period [2]. The period considered to be at risk for VAP ranged from 48 h after intubation until the removal of the tracheal tube and weaning from the invasive ventilation.

#### 2.3.2. Definitions

A relapse was defined as the persistence of clinical symptoms with a new VAP episode due to the same pathogen, after an antibiotic was administered for at least 7 days. A superinfection was defined as a new VAP episode due to a different pathogen that was detected at least 4 days after the first episode. VAP was considered in “treatment failure” if a patient met one of these scenarios: relapse, superinfection, or death from any cause [20]. Mortality from any cause was assessed at Day 60, regardless of whether a patient was still in ICU or had been discharged.

#### 2.3.3. Antimicrobial Resistance

High-risk pathogens were defined as multi-drug resistant (MDR) organisms, which were clustered into four classes: methicillin-resistant *Staphylococcus aureus* (MRSA), extended-spectrum beta-lactamase (ESBL)-producing *Enterobacterales*, AmpC-overexpressing *Enterobacterales*, and *Pseudomonas aeruginosa* resistant to ticarcillin, imipenem, and/or ceftazidime. Carbapenemase-producing *Enterobacterales* (CPE), as well as carbapenem-resistant and carbapenemase-producing non-fermenting Gram-negative bacteria (NF-GNB) (*Acinetobacter baumanii*, *Pseudomonas aeruginosa*, and *Stenotrophomonas maltophilia*), were considered difficult-to-treat pathogens.

In each center, the patients had a systematic screening for ESBL *Enterobacterales* at ICU admission and at least once a week [21,22]. No systematic screening for other MDR organism carriage was performed. A patient was considered colonized if one of these microorganisms was isolated from their screening sample (perirectal area, nose, any screening sample, or any sample performed because of clinical symptoms).

### 2.4. Data Collection

All data were prospectively collected. They comprised details on ICU admission, demographic characteristics, chronic disease, relevant comorbidities, and baseline severity indices, including SAPS II and SOFA score.

Treatments on admission were recorded, including corticosteroids, IL-6 or IL-1 receptor antagonists, and previous antibiotic treatments. Several variables were recorded throughout the ICU stay, including clinical and biological parameters; requirements for non-invasive ventilatory support and invasive mechanical ventilation (IMV); prone position; the presence of a pulmonary embolism; other organ support (vasopressors, renal replacement therapy (RRT), and ECMO); and antimicrobial therapy. Biological analyses were performed as clinically indicated. The presence of glomerular hyperfiltration was usually evaluated but not always available (defined by a glomerular filtration rate above 130 mL/min/m^2^).

Outcomes measures that were routinely recorded were the occurrence of VAP; ICU and hospital length of stay (LOS); and vital status when in ICU, at hospital discharge, and at Day 60 after ICU admission. The duration of treatment for each antibiotic, either mono- or bi-therapy, was collected for each episode of infection.

### 2.5. Case Management

Within the OutcomeRea™ network, the strategies for ventilation modalities, timing for intubation, and other supportive care are decided at each local level according to current guidelines. To confirm pneumonia, the choice of quantitative sampling methods is at the discretion of the attending physicians (endotracheal aspirates, distal plugged catheter, and bronchoalveolar lavage).

This pragmatic approach is routinely performed in all participating centers of the OutcomeRea™ Network. Empirical antimicrobial utilization is decided by the physicians of each center according to international guidelines and local epidemiology [23,24], as it is for de-escalation of antimicrobial therapy. The planned duration of antibiotic therapy for VAP is 7 days according to guidelines, but can be prolonged on an individual basis based on clinical status [25].

### 2.6. Statistical Analysis

The characteristics of the patients were expressed as number (percentage) for categorical variables and median [interquartile range (IQR)] for continuous variables. Comparisons were performed for continuous and categorical variables using Pearson’s Chi-Squared test or exact Fisher’s test, and Wilcoxon’s rank sum test, as appropriate.

Our statistical plan had two steps. First, we assessed the impact of VAP on 60-day mortality using a time-dependent Cox model. Since the risk factors for death in COVID-19 patients were not fully elucidated, and due to the known multicollinearity between several of the selected factors, we used a random forest method to identify significant explanatory variables for inclusion in the multivariate model. A random forest method is a type of decision tree learning algorithm that is able to address nonlinear relationships and complex interactions between potential explanatory variables and rank the relative importance of each factor in the prediction of the occurrence of death. The random forest analysis was conducted using the package randomForest (CRAN; version 4.6–14) in R (CRAN; version 3.6.3). The ranking of the first 20 preselected factors was then used as the order in which the factors were added to the survival model using a forward variable selection based on the Bayesian information criterion (BIC). The factors identified as important during the forward variable selection were then further explored using a backward selection survival model, in order to determine their association with 60-day mortality. All terms with a *p*-value of <0.05 remained in the backward selection survival model. A hazard ratio (HR) >1 indicated an increased risk of death. The proportionality of hazard risks for the covariates was assessed using the marginal residuals.

Second, subgroup analyses among the patients with VAP were conducted to assess the impact of relapse, superinfection, and treatment failure on 60-day mortality, using similarly selected covariates.

Third, the risk of treatment failure (relapse, superinfection, or death, depending on which event occurred first) at Day 60 was computed using a univariate and a multivariate survival model and covariate selection using the random forest method. In order to identify the risk factors for relapse, superinfection, and treatment failure among the patients with VAP, we performed multivariable Fine and Gray sub-distribution competing risk models. ICU death and extubation were considered competing events. The variables associated with these events with a *p*-value of <0.2 were introduced in the multivariate model. A backward selection was performed to select the final covariates.

All models were stratified by center. For all tests, a two-sided α of 0.05 was considered significant. Missing baseline variables were handled through multiple imputation with only one dataset. All statistical analyses were performed using the SAS software, Version 9.4 (SAS Institute, Cary, NC, USA), and R (version 3.6.3).

### 2.7. Ethics

The methods for data collection and the quality of the database have been described previously. In accordance with French laws, the OutcomeRea^TM^ database has been approved by the French Advisory Committee for Data Processing in Health Research (CCTIRS) and the French Informatics and Liberty Commission (CNIL, registration no. 8999262). The database protocol was submitted to the Institutional Review Board of the Clermont-Ferrand University Hospital (Clermont-Ferrand, France), who waived the need for informed consent (IRB no. 5891). The analysis of the risk factors and the outcome associated with VAP relapse was approved by the ethical committee of the French Society of Intensive Care (# SRLF 21-76)**.**

## 3. Results

### 3.1. Main Characteristics and Comparison between Patients with and without VAP

Of the 1424 patients who were admitted in our ICUs during the study period with COVID-19 acute respiratory failure and were potentially eligible for inclusion, 884 patients were excluded because they were mechanically ventilated for less than 48 h. Finally, 540 patients were included; among these patients, 229 presented at least one episode of VAP, and 311 had no diagnosis of VAP during their stay (Flow diagram, Appendix A). Their underlying conditions and main characteristics at admission to an ICU are displayed in Table 1.

The patients with VAP were roughly similar to other invasively ventilated patients upon ICU admission. Corticosteroids were prescribed for about one half of the patients in admission, more frequently in those who would subsequently develop VAP. VAP was more frequent in the patients with ARDS requiring more than 12 cm H_2_O positive end-expiratory pressure (PEEP) during the first 48 h of mechanical ventilation, and those who needed prone position and ECMO (Table 1). Previous use of antimicrobials (e.g., amoxicillin-clavulanate and ureido-carboxypenicillins) before the period at risk (i.e., from ICU admission to 48 h after invasive mechanical ventilation started) was more frequent in the patients who subsequently developed VAP.

### 3.2. Microbiological Results

The pathogens recovered from the samples for VAP diagnostics are detailed in Appendix A. *Enterobacterales* were the causative pathogens for 49.8% of those episodes, followed by *P. aeruginosa* (24.8%) and *S. aureus* (22%). Relapses and superinfections never occurred after *Streptococcus pneumoniae*, *Moraxella catarrhalis*, or *Haemophilus influenzae* infections. The most common pathogens responsible for relapse and superinfection were *Enterobacter* spp., which occurred in 30.5% and 20.5% of cases, respectively; *P. aeruginosa*, which occurred in 35.2% and 34.2% of cases, respectively; and *S. aureus*, which occurred in 22.9% and 27.4% of cases, respectively. More than one strain was isolated in relapse and superinfection in 33.3% and 42.5% of the patients, respectively.

### 3.3. VAP Incidence and Impact on Mortality

The cumulative incidence of VAP at Day 30 and Day 60 was 60% (95% CI [59.8–60.1]) and 61.5% (95% CI [61.3–61.7]), respectively (Figure 1). The incidence rate was 45.6/1000 ventilator days. Among 231 patients with VAP, 79.2% had at least one late-onset episode. The length of stay in an ICU was twice as high for the patients diagnosed with VAP. The median number of ventilator-free days at Day 60 was zero [0–36] in the patients with VAP. The crude 60-day fatality rate was not significantly higher among the patients with VAP compared to those without (47.6% vs. 46.3%, *p* = 0.49), and a higher proportion of these patients were still in an ICU at Day 60 (Table 2).

The variables that are univariately associated with 60-day mortality are presented in Appendix A. In a multivariate survival analysis, we found that age, immunosuppression, cardiovascular comorbidities, lymphocytes/neutrophils ratio > 0.1, parenteral feeding, and VAP were associated with higher mortality (Table 3). The main and subgroup analyses showed that VAP increased mortality (HR 1.36; 95% CI [1.03–1.8]). The association was only significant for late-onset episodes (HR 1.56; 95% CI [1.14–2.13]). Relapse and superinfection were not associated with an increased risk of death in the 229 patients with at least one episode (Table 4).

### 3.4. Risk Factors Associated with Treatment Failure

The cumulative incidences of relapse, superinfection, and treatment failure at Day 30 were 30.9% (95% CI [30.6–31.2]), 28.9% (95% CI [28.6–29.2]), and 42.7% (95% CI [42.4–43]), respectively. The incidences of relapse, superinfection, and treatment failure increased until Day 60, up to 45% (95% CI [44.5–45.6]), 39.5% (95% CI [36.9–37.8]), and 58.1% (95% CI [57.6–58.6]), respectively (Figure 1).

Using a Fine and Gray model, the factors associated with superinfection were an ICU admission prior to May 2020 (HR 0.40, 95% CI [0.22–0.71]), ECMO requirement at intubation (HR 2.13, 95% CI [1.13–4.00]), and first VAP due to NF-GNB (HR 0.37, 95% CI [0.17–0.79]). Age > 70 years old (HR 1.91, 95% CI [1.36–2.68]), corticosteroid use (HR 1.54, 95% CI [1.09–2.16]), and the need for RRT at the time of VAP (HR 1.43, 95% CI [1.06–1.93]) were significantly associated with treatment failure (Table 5). Bi-antimicrobial therapy and broad-spectrum therapy were not associated with the risk of relapse or treatment failure in the univariate analysis (Appendix A).

## 4. Discussion

### 4.1. Impact of Case Severity on VAP and Mortality

In our multicentric cohort, we confirmed the peculiar high rate of VAP in COVID-19 mechanically ventilated patients, with an estimated cumulative incidence as high as 61.5%. It occurred more frequently in the patients with multiple organ failures and with severe ARF (high PEEP level, use of prone position, and ECMO during the first two days of mechanical ventilation) [6]. The initial use of broad-spectrum antibiotics upon ICU admission was associated with mortality in the univariate analysis and was also associated with an increased risk of VAP. These results argue against an extensive use of empirical antimicrobial therapy in COVID-19 patients with acute respiratory failure, where the risk of bacterial co-infection is rare [13].

The 60-day mortality rate of 47.6% was also particularly high, compared to previous cohorts (Appendix A). VAP implied a duration of mechanical ventilation and an ICU length of stay that was prolonged by twofold. It was logically associated with an estimated 36% increase in the risk of death, after a careful adjustment for other confounding prognostic covariates. About three-fourth of the episodes were late-onset VAP episodes, compared to about 50% among other mechanically ventilated ARDS patients [26]. Therefore, the higher risk of death is likely explained by the long duration of mechanical ventilation, which increases the risk of late-onset pneumonia [27,28]. Indeed, the impact of early-onset pneumonia on mortality was negligible, while that of late-onset pneumonia reached 56% in our cohort.

The over-risk of death after VAP has rarely been estimated in ventilated COVID-19 patients, but this fraction could represent as much as 86% [29]. This suggests a need for assessing the respective impact of the severity of ARF, mechanical ventilation, and SARS-CoV-2 infection itself. The available studies showed a maximal increase in mortality when the severity of organ failures is intermediate [27,28]. The level of oxygenation failure and the severity of previous ARF may also modify the over-risk of death in SARS-CoV-2 ARF [30]. Some authors concluded that VAP had less impact on in-hospital mortality, while mainly delaying extubation and discharge [31,32]. One analysis of the French REA-REZO network suggested that the attributable fraction of mortality due to VAP was significantly higher for COVID-19 patients than for other patients [2]. This was explained by both a decreased risk of death in the ventilated patients who did not acquire VAP and an increase in case of VAP acquisition.

### 4.2. Impact of Organ Support and Pharmacokinetic Challenges

In our cohort, ECMO was needed initially for 13.5% of the ventilated patients, and up to one-fourth of the patients thereafter. When required at VAP onset, ECMO was associated with a more than 2-fold increase in the risk of superinfection. Our results are consistent with previous studies on VAP among ECMO patients. In non-COVID ARDS patients, relapses have been reported in up to 30% of cases, with a 3-fold increase in mortality [33,34]. A recent study showed that relapses and superinfections occurred in 79% of COVID-19 patients assisted with ECMO [15]. The challenge of a VAP diagnosis in this population may have induced misdiagnoses and delayed appropriate therapy [5,35]. Thus, it may have increased the burden of VAP, while favoring the risk for treatment failure.

Renal replacement therapy had a substantial impact. It significantly increased the risk of treatment failure by 43% (*p* = 0.02) and marginally increased the risk of death by 44% (*p* = 0.06). Unfortunately, the role of potential underdosage of antibiotics or pharmacokinetic alterations (i.e., altered tissue perfusion, increased distribution volume, and augmented renal clearance) [36] has not been investigated. Yet, it may explain the high number of treatment failures and justify the need for further studies. Indeed, multi-organ dysfunction has been associated with lung abscesses after unresolved VAP, seen in 14% of critically ill ventilated COVID-19 patients [17]. Our results assert the need for antibiotic combination and dosage optimization, given the high risk of failure.

### 4.3. High Incidence of Treatment Failure and Clinical Implications

In our cohort, relapse and superinfection had a cumulative incidence of 45% and 39.5%, respectively, which is particularly high compared to other studies [1,2,4]. Surprisingly, the incidence of treatment failure (relapse, superinfection, or death) at Day 60 reached 58.1%, although without adding to the over-risk of death. To explain this, we could hypothesize a combination of immunological and microbiological factors, along with therapeutic issues.

First, the extended use of corticosteroids has presumably enhanced the local and systemic SARS-CoV-2-related immune alterations, although controversies exist upon their deleterious impact in terms of superinfections [37,38]. We found an association between steroid use and both the risk of VAP and 60-day mortality. Additionally, the risk of treatment failure increased by more than 50% for the patients receiving steroids. These results suggest a deleterious impact of steroid therapy as a treatment for SARS-CoV-2 infection associated with acute respiratory failure. Indeed, the most severely ill subgroup of patients, who are mechanically ventilated for a long period of time and, thus, at risk of multiple infectious complications, might not benefit from it.

Second, the bacterial etiologies of VAP provide relevant explanation. Relapses occurred more frequently after VAP caused by non-fermenting species (NF-GNB), group 3 Enterobacterales, and *S. aureus*. In non-COVID patients under ECMO, VAP due to NF-GNB was associated with a 2-fold increase in the risk of treatment failures [34]. In COVID-19 patients, *P. aeruginosa* and *A. baumanii* were twice more frequent in relapses than in first episodes, reaching more than 40%, whereas ESBL-producing Enterobacterales were recovered for only 15% to 25% of relapses [16,39]. The high risk of late-onset pneumonia due to non-fermenting species may also be related to the extensive use of broad-spectrum antibiotic therapies. It may explain why their use adversely affects 60-day mortality.

In the univariate analysis, we found that bi-antimicrobial therapy for the first episode of VAP significantly decreased the risk of superinfection (Appendix A). The superiority of an active combination therapy over a monotherapy still raises debate. For VAP due to difficult-to-treat non-fermenting species, a dual active therapy could improve microbiological eradication [40,41]. However, recent pivotal studies failed to demonstrate a true benefit of an initial use of combination therapy in terms of clinical cure or mortality [42,43]. Considering the high rate of late-onset VAP due to difficult-to-treat microorganisms and pharmacokinetic alterations associated with severe COVID-19 ARDS, the optimization of initial antimicrobial doses and therapeutic drug monitoring should be promoted. A combination therapy should be discussed on an individual basis.

### 4.4. Limits

Our study had some limitations. The diagnosis of VAP was based on similar definitions and required a confirmation from an assessment of quantitative bacterial culture. However, we could not be sure that the criteria for suspected VAP were similar across centers, especially for patients with diffuse infiltrates and/or under ECMO. However, the stratification by center limited the impact of information bias. Still, the period of admission might have influenced the standard of care, in particular regarding corticosteroid use and ventilation strategies during the first wave. For instance, the timing and indications for mechanical ventilation and the availability of ECMO have evolved over time. In addition, the infection prevention and control measures might have been less rigorously followed due to staff overload, thus explaining an increased risk of nosocomial infections. Eventually, the not-to-intubate decision could have been made in some situations in highly overcrowded settings. This might have substantially affected the incidence of VAP and the estimation of the increase in the risk of death. Finally, the impact of new variants also remains an open issue.

## 5. Conclusions

In our multicentric cohort of COVID-19 patients, the cumulative incidence of VAP was singularly high, reaching 61.5% at Day 60. It was associated with an increased attributable mortality, with late-onset episodes having the highest impact on the mortality hazard. Our study also showed a higher risk of treatment failure, which was potentially explained by difficult-to-treat microorganisms and which increased after corticosteroid use, pharmacokinetic alterations due to organ failure, renal replacement therapy, and ECMO. Relapses and superinfections were not responsible for an over-risk of mortality but implied the need for an individualized approach. The benefit of a combination therapy and the duration of treatment should be evaluated in future research.

## Figures and Tables

**Figure 1 jcm-12-01298-f001:**
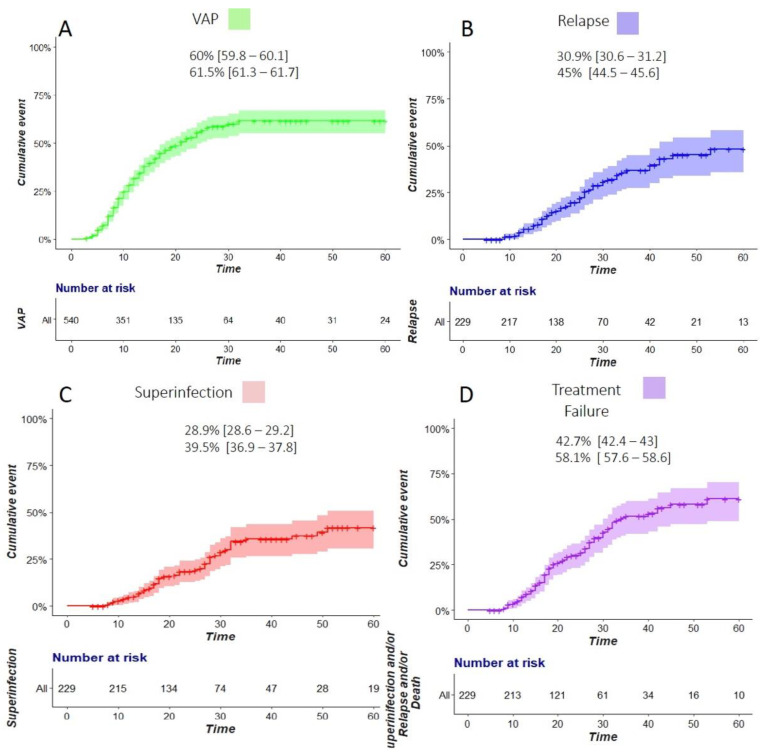
Cumulative incidence of ventilator-associated pneumonia (VAP) (**A**), relapse (**B**), superinfection (**C**), and treatment failure (**D**). The estimations at Day 30 and Day 60 are displayed for each respective outcome (% [95% confidence interval]).

**Table 1 jcm-12-01298-t001:** Characteristics of patients at risk of ventilator-associated pneumonia (VAP).

	All n = 540	No VAP n = 311	VAP n = 229	*p*-Value
Period (before May 2020)	284 (52.6)	162 (52.1)	122 (53.3)	0.79
Age	63.6 [54.6–71.8]	65.2 [55.8–72.6]	62.3 [52.5–71]	0.02
Gender (Male)	401 (74.2)	223 (71.7)	178 (77.7)	0.11
Body mass index	28.8 [25.4–32.6]	28.4 [25–32.1]	29.4 [26.2–33.6]	<0.01
At least one comorbidity	357 (66.2)	215 (69.1)	142 (62)	0.08
Chronic liver failure	11 (2)	7 (2.3)	4 (1.7)	0.68
Chronic cardiovascular disease	151 (28)	99 (31.8)	52 (22.7)	0.02
Chronic respiratory failure	60 (11.2)	36 (11.6)	24 (10.5)	0.69
Chronic kidney disease	48 (8.8)	33 (10.6)	15 (6.6)	0.10
Immunosuppression ^¶^	58 (10.8)	44 (14.1)	14 (6.1)	<0.01
Diabetes mellitus	98 (18.2)	57 (18.3)	41 (17.9)	0.90
Characteristics during ICU admission				
Time from symptom onset (miss = 54)	9 [7–12]	9 [6–12]	9 [7–12]	0.67
SAPS II score	38 [29–51]	39 [31–53]	38 [29–48]	0.09
COVID-19 specific treatments during admission				
Corticosteroids	277 (51.2)	148 (47.6)	129 (56.3)	0.04
Dexamethasone ^†^	222 (41.2)	114 (36.7)	108 (47.4)	0.01
High dose	70 (13)	36 (11.6)	34 (14.9)	0.25
Low dose	158 (28.2)	78 (25.1)	80 (32.3)	0.08
Hemisuccinate hydrocortisone	32 (6)	19 (6.1)	13 (5.7)	0.84
Methylprednisolone	6 (1.2)	4 (1.3)	2 (0.9)	0.65
Prednisone	12 (2.2)	7 (2.3)	5 (2.2)	0.96
Tocilizumab	24 (4.4)	13 (4.2)	11 (4.8)	0.72
Organ supports before the period at risk *				
PEEP > 12 cm H_2_O	166 (30.8)	81 (26)	85 (37.1)	<0.01
Prone position	175 (32.4)	84 (27)	91 (39.7)	<0.01
Neuromuscular blockade	436 (80.8)	244 (78.5)	192 (83.8)	0.12
ECMO	47 (8.8)	16 (5.1)	31 (13.5)	<0.01
Renal Replacement Therapy	56 (10.4)	35 (11.3)	21 (9.2)	0.43
Vasopressor	306 (56.6)	193 (62.1)	113 (49.3)	<0.01
Enteral feeding	332 (61.4)	183 (58.8)	149 (65.1)	0.14
Parenteral feeding	106 (19.6)	64 (20.6)	42 (18.3)	0.52
Proton pump inhibitor	314 (58.2)	178 (57.2)	136 (59.4)	0.62
Organ supports during ICU stay				
Prone position	284 (52.6)	123 (39.5)	161 (70.3)	<0.01
ECMO	80 (14.8)	26 (8.4)	54 (23.6)	<0.01
Renal replacement therapy	178 (33)	91 (29.3)	87 (38)	0.03
Vasopressor	362 (67)	215 (69.1)	147 (64.2)	0.23
Before the period at risk *				
At least one antimicrobial therapy				
Amoxicillin/clavulanic acid	64 (11.8)	29 (9.3)	35 (15.3)	0.03
Ureido-carboxypenicillins	91 (16.8)	42 (13.5)	49 (21.4)	0.02
3rd-generation cephalosporin	316 (58.6)	189 (60.8)	127 (55.5)	0.22
4th-generation cephalosporin	89 (16.4)	50 (16.1)	39 (17)	0.77
Carbapenem	40 (7.4)	28 (9)	12 (5.2)	0.10
Macrolide	193 (35.8)	111 (35.7)	82 (35.8)	0.98
Fluoroquinolone	64 (11.8)	34 (10.9)	30 (13.1)	0.44
MDR pathogen colonization	45 (8.4)	20 (6.4)	25 (10.9)	0.06
ESBL-producing *Enterobacterales*	34 (6.2)	17 (5.5)	17 (7.4)	0.35
Carbapenem-resistant *Enterobacterales*	7 (1.2)	1 (0.3)	6 (2.6)	0.02
MDR *Pseudomonas aeruginosa*	2 (0.4)	1 (0.3)	1 (0.4)	0.83
MRSA	4 (0.8)	2 (0.6)	2 (0.9)	0.76

Qualitative data were reported as *n* (%) and quantitative data were reported as median [25th percentile-75th percentile] as appropriate. * The period at risk for VAP ranges from 48 h after IMV until weaning from invasive ventilation; ^¶^ leukocytes < 1000 µL^−1^, neutrophils < 500 µL^−1^, acquired or congenital immunodeficiency syndrome, and use of immunosuppressants or long-term corticosteroids (≥0.5 mg.kg^−1^.day^−1^). ^†^ High dose: 20 mg per day intravenously. Low dose: 6 mg per day intravenously. ICU: intensive care unit; SAPS II: simplified acute physiology score; IMV: invasive mechanical ventilation; PEEP: positive end-expiratory pressure; ECMO: extracorporeal membrane oxygenation; MDR: multi-drug resistant; ESBL: extended-spectrum ß-lactamase; MRSA: methicillin-resistant *Staphylococcus aureus*; VAP: ventilator-associated pneumonia.

**Table 2 jcm-12-01298-t002:** Main outcomes according to the occurrence of ventilator-associated pneumonia (VAP).

	All n = 540	No VAP n = 311	VAP n = 229	*p*-Value
At least one episode of early-onset VAP	93 (17.2)	-	93 (40.6)	-
At least one episode of late-onset VAP	179 (33.2)	-	179 (78.2)	-
1 episode of VAP	143 (26.4)	-	143 (62.4)	-
2 episodes of VAP	53 (9.8)	-	53 (23.1)	-
≥3 episodes of VAP	33 (6.2)	-	33 (14.4)	-
At least one superinfection of VAP ^§^	58 (10.8)	-	58 (25.3)	-
1 superinfection of VAP	46 (8.6)	-	46 (20.1)	-
≥2 superinfection of VAP	12 (2.2)	-	11 (4.7)	-
At least one episode of relapse ^‡^	62 (11.4)	-	62 (27.1)	-
1 relapse episode of VAP	39 (7.2)	-	39 (17)	-
≥2 relapse episode of VAP	23 (4.2)	-	22 (9.6)	-
Invasive mechanical ventilation duration *	13 [7–23.6]	9 [5–14]	21 [14–34]	<0.01
VFD at Day 60 *	10 [0–47]	29 [0–52]	0 [0–36]	<0.01
ICU LOS *	16 [10–29]	13 [8–19]	26 [17–41]	<0.01
Hospital LOS *	22 [13.6–40]	17 [10–30]	31 [19–50]	<0.01
ICU death	248 (46)	138 (44.7)	110 (47.6)	0.49
Death at Day 60	263 (48.8)	143 (46.3)	110 (47.6)	0.76
Number of patients still in ICU at Day 60	18 (3.4)	3 (1)	15 (6.5)	<0.01

VAP: ventilator-associated pneumonia; VFD: ventilator-free days; ICU: intensive care unit; LOS: length of stay. ^§^ different pathogen than previously isolated; ^‡^ same pathogens as previously isolated; * starting from the period at risk (≥48 h invasive ventilation).

**Table 3 jcm-12-01298-t003:** Risk factors associated with 60-day mortality, using a Cox’s survival model (multivariate analysis).

	Hazard Ratio	HR 95% CI	*p*-Value
Age			
<50	1	-	-
50–60	1.54	[0.87–2.7]	0.14
60–70	2.88	[1.68–4.94]	<0.01
>70	3.69	[2.15–6.25]	<0.01
Cardiovascular comorbidities	1.43	[1.1–1.87]	<0.01
Immunosuppression	1.76	[1.25–2.49]	<0.01
Renal replacement therapy *	1.44	[0.99–2.10]	0.06
Parenteral feeding	1.51	[1.06–2.17]	0.02
Lymphocyte/neutrophil ratio > 0.1	0.68	[0.49–0.94]	0.02
Ventilator-associated pneumonia	1.36	[1.03–1.8]	0.03

* before the period at risk for ventilator-associated pneumonia (VAP). The variables associated with death that were selected by random forest and tested in multivariate analyses are the following: period of admission, age, gender, BMI, cardiovascular comorbidities, immunosuppression, immunomodulatory treatment upon admission, steroids upon admission, pneumonia upon admission, late intubation, parenteral feeding, vasopressors before VAP, renal replacement therapy before VAP, ECMO before VAP, lymphocyte/neutrophil ratio, bi-antimicrobial therapy upon admission, antibiotics before VAP, broad-spectrum antibiotic therapy before VAP, time (%) with ATB before VAP, and broad-spectrum antibiotic therapy the day of VAP.

**Table 4 jcm-12-01298-t004:** Risk factors associated with 60-day mortality, using different survival models (multivariate analysis).

		Hazard Ratio	HR IC 95%	*p*-Value
Model 1	VAP	1.36	[1.03–1.8]	0.03
Model 2	Early-onset VAP	1.04	[0.77–1.43]	0.78
	Late-onset VAP	1.56	[1.14–2.13]	<0.01
Among 229 patients with at least one episode of VAP				
Model 3	Superinfection	1.04	[0.57–1.9]	0.91
Model 4	Relapse	1.12	[0.64–1.97]	0.70

All models are adjusted on the same variables (see Table 3). VAP: ventilator-associated pneumonia.

**Table 5 jcm-12-01298-t005:** Risks factors associated with superinfection and treatment failure (relapse, superinfection, or death) using a multivariable Fine and Gray competing risk model.

	Hazard Ratio	HR 95% IC	*p*-Value
Superinfection			
Period admission before May 2020	0.40	[0.22–0.71]	<0.01
ECMO *	2.13	[1.13–4.00]	0.02
Non-fermentative bacteria	0.37	[0.17–0.79]	0.01
Treatment failure			
Age > 70 y	1.91	[1.36–2.68]	<0.01
Steroids *	1.54	[1.09–2.16]	0.01
Renal replacement therapy *	1.43	[1.06–1.93]	0.02

* before the period at risk for ventilator-associated pneumonia (VAP). In our survival model, “day 0” is the day of VAP diagnosis. The variables selected by the random forest procedure and tested in the multivariate model are the following: period of admission, age, gender, BMI, immunosuppression, steroids, tocilizumab, late intubation, high-level PEEP, prone position, ECMO, vasopressor use before VAP, renal replacement therapy before VAP, proton pump inhibitor, highly susceptible germs, group 1–2 *Enterobacterales*, group 3 *Enterobacterales*, *Staphylococcus aureus*, non-fermenting species, MDR or XDR, MDR colonization before VAP, bi-antimicrobial therapy, and broad-spectrum antibiotic therapy the day of VAP.

## Data Availability

Data are available to the corresponding author upon reasonable and scientifically reasonable request.

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
