# Peer review of "Ventilator-Associated Pneumonia in COVID-19 Patients Admitted in Intensive Care Units: Relapse, Therapeutic Failure and Attributable Mortality—A Multicentric Observational Study from the OutcomeRea Network"

_jcm, 2023, doi:10.3390/jcm12041298_

Round 1

Reviewer 1 Report

Dear authors,

In my review, I suggest minor adjustments to the results, discussion, and conclusion to publish this article

In the paragraph on line 305, I think you should add all the results shown in table 5 about risk factors for treatment failure using the multivariate analysis such as  age greater than 70.

In the paragraph, line 392,  I didn’t find the result supporting that affirmation about bi-antimicrobial therapy for the first VAP decreased superinfection.

In addition, I think You should explain more about other risk factors for treatment failure such as age and renal replacement therapy. You must show this result.

The conclusion should be more objective and direct, emphasizing the study's findings, such as the risk factors for mortality in VAP and treatment failure. Another important topic to explain in the conclusion is the use carefully of corticoids in these patients. This issue should be better evaluated since it is a common practice in treatment protocols for patients on mechanical ventilation due to covid.

Author Response

In my review, I suggest minor adjustments to the results, discussion, and conclusion to publish this article.

We thank the reviewer for the comments and suggestions provided, that help improving our manuscript.

  • In the paragraph on line 305, I think you should add all the results shown in table 5 about risk factors for treatment failure using the multivariate analysis such as age greater than 70.

We thank the reviewer for this comment. Corrections have been made in the manuscript.

  • In the paragraph, line 392, I didn’t find the result supporting that affirmation about bi-antimicrobial therapy for the first VAP decreased superinfection.

We thank the reviewer for this remark. Indeed, those results have been reported in the Supplement Table 3. Subsequent corrections have been provided in the manuscript.

  • In addition, I think You should explain more about other risk factors for treatment failure such as age and renal replacement therapy. You must show this result.

We are grateful for this suggestion. We added those results (Line 308-309). Also, we explain the results in the discussion, referring to tables 3 and 5, in the paragraph about organ support (Line 362).

  • The conclusion should be more objective and direct, emphasizing the study's findings, such as the risk factors for mortality in VAP and treatment failure. Another important topic to explain in the conclusion is the use carefully of corticoids in these patients. This issue should be better evaluated since it is a common practice in treatment protocols for patients on mechanical ventilation due to covid.

We are thankful for this comment, about those two critical issues. Indeed, the risk of infectious complications associated with the wide use of corticosteroids was debated in the literature, and we mention this point in the discussion and limits sections, line 381 and line 415. We made corrections concerning both aspects in the conclusion of the manuscript.

Reviewer 2 Report

Dear Authors,

1. The title should not consist abbreviations.

2. The conclusion should not have a part of a discussion - it can be improve.

The overall merit for the paper is high.

Author Response

  1. The title should not consist abbreviations- agree, now modified
  2. The conclusion should not have a part of a discussion - it can be improved. We thank the reviewer for those comments and suggestions. Corrections have been made in the manuscript.

The overall merit for the paper is high. We are grateful to the reviewer for this comment.

Reviewer 3 Report

Dear Editor, 

Thank you for the opportunity to review the study by Wicky et al on VAP in Covid-19 ICU patients. The study evaluated the mortality associated with VAP in mechanically ventilated Covid-19 patients in a large french database. It is a very interesting study, with important results. 

However, several clarifications are required to evaluate the results of the study. Specifically:

Lines 89-90: the mechanisms predisposing to VAP should be better presented - apart from the viral-induced immunomodulation, the other mechanisms mentioned are either not valid or unique to Covid

Line 91: ‘added value’ rephrase

Line 96: ‘some authors’ rephrase

Lines 99-102: please rephrase, this sentence is confusing

Line 133: VAP definition please clarify if only microbiologically confirmed episodes were included - and how other criteria (oxygenation/pressors) were used in the definition

Line 138: how was ‘alteration in oxygenation’ defined

Line 148: how was relapse defined? Persistence of which symptoms?

Line 150: ‘4 days after’ you mean at least 4 for days after?

Line 152: defining treatment failure death irrespective of the cause is problematic and should be mentioned in limitations - if solely death caused by respiratory failure or septic shock due to VAP could be documented that would certainly establish ’treatment failure’

Please include definitions of immunosuppression, and provide details on the type/dose of steroids used, as it changed over time (Line 262)

Data on oxygenation-PEEP 1-2 days prior to the diagnosis of VAP would provide important information about the severity of Covid and the relative contribution of VAP to the prolongation of ICU stay (for example a patient on a PEEP of 5 and FiO2 of 30 would be more likely to be weaned if VAP hadn’t occur, while for a patient on a PEEP of 15 and fiO2 of 60, the impact of a VAP on LOS is at least questionable) - you may want to expand on discussion on this 

Lines 266-8: please rephrase to clarify

Lines 377-78: if there are adequate data it would be important to investigate if the recommended dose of steroids (dexamethasone 6 mg) alone is associated with increased risk of VAP, or if the observed association with increased risk is associated with higher doses/combinations with other immunosuppressants. 

Line 427-8: please omit this statement on pharmacokinetic optimization in conclusion as it is not directly supported by any data from the study

Author Response

Thank you for the opportunity to review the study by Wicky et al on VAP in Covid-19 ICU patients. The study evaluated the mortality associated with VAP in mechanically ventilated Covid-19 patients in a large french database. It is a very interesting study, with important results. 

However, several clarifications are required to evaluate the results of the study. Specifically:

  • Lines 89-90: the mechanisms predisposing to VAP should be better presented - apart from the viral-induced immunomodulation, the other mechanisms mentioned are either not valid or unique to Covid.

We are grateful for this relevant and accurate remark. We concisely presented some mechanisms involved, since it did not seem useful enough to discuss in depth the physiopathological aspects predisposing to VAP. An overview of the main characteristics debated in the mentioned literature would avoid entering a debate that was considered non necessary. The other predisposing factors are discussed in our study.

  • Line 91: ‘added value’ rephrase.

Corrections have been made in the manuscript, to make it clearer.

  • Line 96: ‘some authors’ rephrase

Corrections have been made in the manuscript, to make it clearer.

  • Lines 99-102: please rephrase, this sentence is confusing

Corrections have been in the manuscript, and we hope that it would make more sense.

  • Line 133: VAP definition please clarify if only microbiologically confirmed episodes were included - and how other criteria (oxygenation/pressors) were used in the definition.

We acknowledge the reviewer for this precise question. Each episode had to be microbiologically confirmed to be included. It is important to precise, that a microbiological sample was performed only if clinically relevant. We admitted that a pragmatic approach was commonly privileged across all centers, following the international guidelines provided by the European Respiratory Society (ERS/ESICM/ECMID/ALAT) in the paper by Torres et al. (Eur Respir J, 2017, reference #23) to help define Ventilator-Associated-Pneumonia. A VAP was thus expected to be clinically suspected by the physician when an increase in FiO2 (or decrease in the ratio PaO2 / FiO2) or need for vasopressor was observed, without any predefined threshold, and based upon hyperleukocytosis, purulent tracheal secretion, and the subjective interpretation of chest radiograph. Those situations led to pulmonary sample, without a predefined method (either fibroscopy with bronchoalveolar lavage or distal plugged catheter or endotracheal aspirates).

  • Line 138: how was ‘alteration in oxygenation’ defined? As we mentioned above, the increase in FiO2 (or decrease in the ratio PaO2 / FiO2) was considered as alteration in oxygenation.

  • Line 148: how was relapse defined? Persistence of which symptoms?

We acknowledge the reviewer for underlying this point. A relapse was defined as the suspicion of a new episode of VAP based on the same aspects as mentioned before (VAP definition), and when the causing pathogen was the same as previous episode. The symptoms were the same as those leading to the suspicion of VAP (alteration in oxygenation, need for vasopressor).

  • Line 150: ‘4 days after’ you mean at least 4 for days after?

Yes, an interval of at least 4 days after VAP episode.

  • Line 152: defining treatment failure death irrespective of the cause is problematic and should be mentioned in limitations - if solely death caused by respiratory failure or septic shock due to VAP could be documented that would certainly establish ’treatment failure’.

We acknowledge the reviewer for underlying this critical point. Indeed, it was barely possible to confirm that a pejorative evolution of VAP would lead to death. The relationship is sometimes not so obvious. To avoid misdiagnosis in VAP treatment failure when the patient died under treatment, we admitted that death from any cause was acceptable. A pragmatic approach is preferable, and so, we did not consider the precise cause. Consensual clinical endpoints provided in the paper by Weiss et al (Clin Infect Dis, 2019, reference #19) supports this point. To clarify this point, we provided corrections in the manuscript (line 155).

  • Please include definitions of immunosuppression:

The traditional characteristics associated with an existing immunosuppression are those commonly used in studies about sepsis, and mentioned in the guidelines (Torres et al, Eur Respir J, 2017, reference #23), as follow: Leukocytes <1000µL-1, neutrophils <500 µL-1, acquired or congenital immunodeficiency syndrome, use of immunosuppressants or long-term corticosteroids (≥0.5mg.kg-1.day-1). We added these precisions directly in the legend under Table 1 in the manuscript.

  • and provide details on the type/dose of steroids used, as it changed over time (Line 262).

We acknowledge the reviewer for this relevant remark. Corrections have been made in the manuscript, in Table 1, with some precisions about the dose and type of steroid used. Nonetheless, no added results could come out from our analysis.

  • Data on oxygenation-PEEP 1-2 days prior to the diagnosis of VAP would provide important information about the severity of Covid and the relative contribution of VAP to the prolongation of ICU stay (for example a patient on a PEEP of 5 and FiO2 of 30 would be more likely to be weaned if VAP hadn’t occur, while for a patient on a PEEP of 15 and fiO2 of 60, the impact of a VAP on LOS is at least questionable) - you may want to expand on discussion on this. We acknowledge the reviewer for underlying this point.

Indeed, the modifications after the start of period at risk could be either the mirror of a new VAP on incubation or a mirror of deterioration of the Covid ARF. It is typically a Collider in a causal relationship. That is why we propose to only insert data about peep level just before the period at risk (see table 1). However, we agree with the reviewer that the attributable mortality may vary according to the level of oxygenation failure and we added it in the discussion section. 

  • Lines 266-8: please rephrase to clarify

Corrections have been made in the manuscript, and we hope that it would avoid any confusion.

  • Lines 377-78: if there are adequate data it would be important to investigate if the recommended dose of steroids (dexamethasone 6 mg) alone is associated with increased risk of VAP, or if the observed association with increased risk is associated with higher doses/combinations with other immunosuppressants. 

We acknowledge the reviewer for this very interesting remark. Unfortunately, our dataset did not allow us to focus on this particular aspect. The strategy concerning the use of steroids was initially not homogenous between participating centers. Moreover, some of our patients were included in other randomized trial assessing high (20mg) versus low dose (recommended 6mg). So, our study was not powerful enough to confirm a dose-effect with corticosteroids. Also, it was not possible to assess precisely the impact of immunomodulator administration.

  • Line 427-8: please omit this statement on pharmacokinetic optimization in conclusion as it is not directly supported by any data from the study.

We acknowledge the reviewer for this remark. Corrections have been made in the manuscript.

Round 2

Reviewer 3 Report

I have no further comments